# Risk Factors for Hepatitis E Virus Infection and Eating Habits in Kidney Transplant Recipients

**DOI:** 10.3390/pathogens12060850

**Published:** 2023-06-20

**Authors:** Eva Wu, Nadine Koch, Friederike Bachmann, Marten Schulz, Evelyn Seelow, Ulrike Weber, Johannes Waiser, Fabian Halleck, Mirko Faber, Claus-Thomas Bock, Kai-Uwe Eckardt, Klemens Budde, Jörg Hofmann, Peter Nickel, Mira Choi

**Affiliations:** 1Department of Nephrology and Medical Intensive Care, Charité-Universitätsmedizin Berlin, Corporate Member of Freie Universität Berlin and Humboldt-Universität zu Berlin, 13353 Berlin, Germany; eva.wu@charite.de (E.W.); nadine.koch@charite.de (N.K.); friederike.bachmann@charite.de (F.B.); evelyn.seelow@charite.de (E.S.); ulrike.weber@charite.de (U.W.); johannes.waiser@charite.de (J.W.); fabian.halleck@charite.de (F.H.); kai-uwe.eckardt@charite.de (K.-U.E.); klemens.budde@charite.de (K.B.); peter.nickel@charite.de (P.N.); 2Department of Hepatology and Gastroenterology, Charité-Universitätsmedizin Berlin, Corporate Member of Freie Universität Berlin and Humboldt-Universität zu Berlin, 13353 Berlin, Germany; marten.schulz@charite.de; 3Department of Infectious Disease Epidemiology, Robert Koch-Institute, 13353 Berlin, Germany; faberm@rki.de; 4Department of Infectious Diseases, Robert Koch-Institute, 13353 Berlin, Germany; bockc@rki.de; 5Institute of Virology, Charité-Universitätsmedizin Berlin, Corporate Member of Freie Universität Berlin and Humboldt-Universität zu Berlin, Berlin Institute of Health, and German Centre for Infection Research (DZIF), Partner Site Charité, 13353 Berlin, Germany; joerg.hofmann@laborberlin.com; 6Labor Berlin, Charité-Vivantes GmbH, 13353 Berlin, Germany

**Keywords:** hepatitis E, kidney transplant, immunosuppression, meat consumption

## Abstract

There is a significant risk for ongoing and treatment-resistant courses of hepatitis E virus (HEV) infection in patients after solid organ transplantation. The aim of this study was to identify risk factors for the development of hepatitis E, including the dietary habits of patients. We conducted a retrospective single-center study with 59 adult kidney and combined kidney transplant recipients who were diagnosed with HEV infection between 2013 and 2020. The outcomes of HEV infections were analyzed during a median follow-up of 4.3 years. Patients were compared with a control cohort of 251 transplant patients with elevated liver enzymes but without evidence of an HEV infection. Patients’ alimentary exposures during the time before disease onset or diagnosis were assessed. Previous intense immunosuppression, especially treatment with high-dose steroids and rituximab, was a significant risk factor to acquire hepatitis E after solid organ transplantation. Only 11 out of 59 (18.6%) patients reached remission without further ribavirin (RBV) treatment. A total of 48 patients were treated with RBV, of which 19 patients (39.6%) had either viral rebounds after the end of treatment or did not reach viral clearance at all. Higher age (>60 years) and a BMI ≤ 20 kg/m^2^ were risk factors for RBV treatment failure. Deterioration in kidney function with a drop in eGFR (*p* = 0.046) and a rise in proteinuria was more common in patients with persistent hepatitis E viremia. HEV infection was associated with the consumption of undercooked pork or pork products prior to infection. Patients also reported processing raw meat with bare hands at home more frequently than the controls. Overall, we showed that the intensity of immunosuppression, higher age, a low BMI and the consumption of undercooked pork meat correlated with the development of hepatitis E.

## 1. Introduction

Although a hepatitis E virus (HEV) infection is usually a self-limiting disease, immunocompromised patients are prone to developing an ongoing infection with an increased risk of significant chronic liver disease, including cirrhosis [1,2,3]. Hepatitis E is mainly associated with genotype 3 infections. More intense immunosuppression and previous treatment with steroids or T- or B-cell-targeting therapies to treat episodes of cellular or humoral rejection episodes after transplantation are known risk factors for persistent hepatitis E viremia [4,5]. The management of HEV infection in patients after solid organ transplantation (SOT) is still challenging and requires a stepwise approach [6]. Despite an increased vigilance for HEV infection after SOT, patients do not comply with precautionary measures frequently, particularly eating habits, especially over time. On the physician side, many HEV infections remain undiagnosed, or, in the case of chronic HEV infection, are diagnosed late. The reduction in immunosuppression remains the first therapy choice. In cases without viral clearance, treatment with ribavirin for a period of 3 to 6 months can be effective but comes with severe, limiting side effects [7,8,9]. Alternative strategies, such as interferon-alpha, are not suitable for transplant patients [10,11,12], and sofosbuvir failed to achieve promising results [13]. HEV infection may occur late and even many years after kidney or combined-kidney transplantation without evidence of transmission from the donor organ [4]. The consumption of raw or undercooked meat, especially pork, is a main risk factor for acute hepatitis E infection in humans [14,15,16,17]. Changes in food habits, such as the consumption and/or handling of uncooked meat, may be a major factor that leads to genotype 3 hepatitis E and may explain late infection times.

The aim of this study was to identify risk factors for HEV infection among solid organ transplant recipients. In addition, we hypothesized that the unawareness of and/or not following appropriate dietary recommendations carry a risk for HEV infection in patients after transplantation.

## 2. Materials and Methods

### 2.1. Patient Population, Clinical and Laboratory Data

All adult kidney and combined-kidney transplant recipients at Charité-Universitätsmedizin Berlin that were diagnosed with HEV infection between January 2013 and December 2020 were included in this retrospective study. Diagnosis of HEV infection was made via laboratory testing based on a medical indication in the case of an (unexplained) increase in liver enzymes (alanine transaminase, aspartate transaminase and/or gamma glutamyltransferase). Using data extraction from our transplant database (Tbase) [18], we detected 310 out of approximately 2400 transplant patients (~13%) from our center with elevated liver enzymes during the observation period, who underwent further laboratory investigation for HEV. Of these, replication of HEV RNA was detected in 59 patients, which in 48 out of 59 patients lasted for at least three months, thus fulfilling the definition of chronic HEV infection [19]. The other 251 patients were HEV RNA negative and served as our control group to analyze the risk factors for HEV infection. Patients with positive HEV IgG and/or IgM serology results were still included in the CG if the HEV PCR was negative.

After a diagnosis of HEV infection, the intensity of immunosuppression was reduced whenever possible. Patients with hepatitis E viremia persisting for approximately three months were treated with ribavirin (RBV) monotherapy for at least 3 months if tolerated. In the case of a drop in viral load of approximately one log level, the decision for treatment was postponed until the next viral load assessment. Remission was defined as the achievement of viral clearance, either without RBV treatment or after the end of treatment in the case of RBV treatment. Sustained virologic response (SVR) was defined as the absence of viremia for at least six months after the end of treatment. RBV treatment success was defined as a viral clearance and SVR after the first treatment round with RBV. RBV treatment failure was defined as ongoing viremia despite RBV intake or a viral rebound after an initial viral clearance. The Chronic Kidney Disease Epidemiology Collaboration (CKD-EPI) formula was used to estimate the glomerular filtration rate (eGFR). Proteinuria was expressed in mg/g creatinine when available, otherwise set equal to mg per day for previous measurements. Basal eGFR and proteinuria were assessed using the eGFR and proteinuria measures six months prior to the diagnostic evaluation. Of note, data from 17 patients were published previously [4]. However, detailed food habits as part of the risk assessment for HEV infection were not analyzed in the previous study. Furthermore, we report a prolonged follow-up, describing long-term courses of hepatitis E and patient and allograft outcomes.

### 2.2. Data Collection of Food Habits

Using a standardized questionnaire [17], alimentary exposures (e.g., meat and meat products) of the KTR were assessed, referring to the time before disease onset or diagnosis. The KTR with no evidence of HEV infection (either aviremic or via the exclusion of elevated liver enzymes) during the same observation period served as the controls.

### 2.3. Serologic and Molecular Diagnostics and Genotyping of Hepatitis E Infection

HEV serology (anti-HEV IgM and anti-HEV IgG antibodies) and genotyping were assessed as described previously [4]. Detection of HEV mutants was performed as described previously [20].

### 2.4. Assessment of Liver Stiffness

For the liver stiffness measurement, patients underwent either transient elastography (Fibroscan^®^, Echosens, Paris, France) or shear wave elastography measurement using a Canon (formerly Toshiba) Aplio 500 US system (Canon Medical systems Corporation, Otawara, Tochigi, Japan). The examination techniques were described elsewhere [21] and were performed in accordance with guidelines [22]. Transient elastography and shear wave elastography are both well-established and reliable techniques for assessing liver stiffness [23]. Since the measurement results of the two methods are not directly comparable, the corresponding liver fibrosis grades were determined. Cut-off values proposed by Ferraioli et al. were applied for the shear wave elastography measurement [24], while for the transient elastography, cut-off values as proposed by Sandrin et al. [25] were used.

### 2.5. Statistical Analyses

Continuous variables were expressed as the mean ± standard deviation (SD) or median and interquartile range (IQR) according to their distribution. Statistical analyses were performed using GraphPad Prism Version 9 and IBM SPSS statistics version 28.0. Measurements were tested for normal/lognormal distributions prior to the analysis. Differences between groups that deviated significantly from Gaussian distribution were assessed using the Mann–Whitney U test (two groups). Correlations were calculated using Spearman’s rank coefficient. The *p*-values less than 0.05 were considered significant. Significant predictors (*p* < 0.1) in the univariable analysis that were considered clinically relevant were fitted into a multivariable logistic regression model. Qualitative outcomes of different cohorts were assessed using the chi-square test with Yates’ correction. All 95% confidence intervals for proportions were calculated using the Wilson procedure with a correction for continuity.

## 3. Results

### 3.1. Baseline Characteristics of KTRs with HEV Infection and Controls

The baseline characteristics of all KTRs are depicted in Table 1.

Patients with HEV infection were predominantly male (46 males and 13 females vs. 141 males and 110 females in the CG, *p* < 0.01). All patients with hepatitis E viremia were infected with the HEV genotype 3 (majority sub-genotype 3c), which is predominant in Central Europe [26]. Between both groups, there were no significant differences in age; underlying kidney disease; co-morbidities; time after transplantation; or type of transplantation, namely, single kidney, combined pancreas–kidney or multi-visceral transplantation; and postmortem transplant or living donation. Furthermore, the baseline allograft function and baseline amount of proteinuria did not differ. Noteworthy, the KTRs with HEV infection had significantly higher ALT and AST levels than patients in the CG (*p* < 0.001). While we observed an increase in proteinuria in both groups during hepatitis E viremia or during the rise of liver enzymes in the CG, an increase in proteinuria ≥300 mg/g creatinine was significantly more frequent in the KTRs with an HEV infection compared with the CG (30% vs. 14% in the CG, *p* < 0.01).

Regarding maintenance immunosuppression, there was a trend toward higher use of the calcineurin inhibitor (CNI) tacrolimus in the KTRs with hepatitis E than in the CG (83% vs. 71%, n.s.), while cyclosporine A was used more frequently in the CG (5.1% vs. 20.7%, *p* < 0.05). Moreover, mycophenolic acid (MPA) was used more frequently, albeit not significantly, in the KTRs with hepatitis E (96.6 vs. 88.4%). The frequencies of triple and dual maintenance immunosuppression were similar in both groups. Triple immunosuppression mainly consisted of a combination of CNI, MPA and low-dose corticosteroid (36 out of 45 in the KTRs with an HEV infection and 160 out of 176 in the CG). Dual maintenance immunosuppression mainly consisted of a combination of CNI and MPA, which was more often than the combination of CNI and corticosteroid or combinations with an mTOR inhibitor or belatacept. Of note, in the univariable analysis, the KTRs with HEV had a higher rate of previously treated rejection episodes (T-cell and B-cell mediated) (*p* = 0.051), including a significantly higher number of previous treatments for antibody-mediated rejections (ABMR) (*p* < 0.01). They also received high-dose corticosteroids (*p* < 0.01) and rituximab (*p* < 0.01) more often (Table 1). On further multivariable analysis, previous needs for high-dose steroids and treatment with rituximab were significantly associated with the occurrence of HEV infection after SOT, with odds ratios of 4.54 (95% CI 1.41–14.60) and 2.56 (95% CI 1.02–6.42), respectively (Table 2). In addition, the use of cyclosporine (CyA) instead of tacrolimus correlated significantly with a lower risk for HEV infection, with an odds ratio of 0.24 (0.07–0.80).

To characterize the KTRs with the intake of CyA in a CNI-based maintenance therapy, we divided all KTRs by intake of CyA vs. not (Appendix A). The use of CyA was not associated with gender, age, comorbidities or type of transplant. We observed a significantly longer time since transplantation in the KTRs with CyA intake (145 ± 97 days vs. 80 ± 75 days, *p* < 0.001). Furthermore, CyA was used less often in immunosuppressive triple-therapy regimens.

Serology at diagnosis of hepatitis E viremia was available in 56 out of 59 patients with HEV infection. A total of 49 out of 56 patients were seropositive for both IgG and IgM, while the remaining 7 patients were seropositive for IgG only. In contrast, the HEV seroprevalence was 32% in the CG (n = 80). Of these, 69 patients were positive for IgG solely, while 11 patients were positive for both IgG and IgM but were aviremic. The time of infection within the CG could not be determined retrospectively since no retained samples were stored. From these cases, we could not exclude cases of chronic HEV infection without recent viremia since no follow-up PCR tests were available. The seroprevalence in our whole cohort was 43.9% (136 out of 310 KTRs).

### 3.2. Risk Factors for Persisting Hepatitis E Viremia with Subsequent RBV Treatment versus Viral Clearance without RBV Treatment

In total, 48 out of 59 (81.4%) KTRs with HEV infection received RBV treatment, while viral clearance without RBV treatment was seen in 11 patients (Table 3).

The change in IS—mainly a reduction in the CNI target level (n = 11), reduction in mycophenolic acid (n = 11) or reduction in a steroid (n = 4)—did not differ between both groups (16 in the KTRs with HEV infection vs. 3 in the CG). We did not find gender, age, co-morbidities, time after transplantation, type of transplant and immunosuppressive regimens to be significant predictors for the failure to achieve viral clearance without medical treatment. While creatinine and eGFR did not differ between both groups, the baseline and peak proteinuria were significantly higher in the KTRs with persistent HEV viremia. Along the same line, the deterioration in eGFR (defined as a decrease in GFR ≥ 5 mL/min; 62.5% vs. 27.3%, *p* < 0.05) occurred significantly more often in the RBV-treated KTRs with persistent viremia compared with the KTRs with remission without RBV treatment. To exclude the deterioration of kidney function, mainly in the KTRs with impaired allograft function, we divided the KTRs by the status of the allograft function (Appendix A) with creatinine ≤ 1.4 mg/dL vs. >1.4 mg/dL. While the baseline proteinuria did not differ between both groups, we observed higher peak proteinuria during the HEV infection in the KTR group with impaired allograft function (*p* = 0.042). In contrast, the percentage of KTRs with an increase in proteinuria (>300 mg/g Crea) during HEV infection was not higher in the KTRs with elevated creatinine (27.2% vs. 23.1%). Furthermore, the numbers of stable eGFR were higher in the KTRs with creatinine ≤ 1.4 mg/dL(50% vs. 39.4%; *p* = 0.441). Thus, we concluded that increases in proteinuria and deterioration in kidney function during HEV infection were not solely a result of chronic kidney damage, but were potentially HEV-associated.

Failure to achieve remission without medical treatment was associated with prolonged viremia (278 ± 203 vs. 126 ± 104 days, *p* < 0.01), with a higher rate of viral rebounds after the initial viral clearance (n = 17 vs. 0, *p* < 0.05). Only 33 out of 48 patients (69%) on the RBV treatment reached an SVR in the long term. The KTRs with persistent virus replication also peaked with a higher maximal viral load (1.8 ± 2.1 vs. 0.7 ± 1.5 mio cop/mL, *p* = 0.01).

The measurement of liver stiffness using transient elastography or shear wave elastography was performed in 32 KTRs with hepatitis E (Table 3). Out of the 28 KTRs treated with RBV, moderate to severe liver fibrosis was observed in 15 patients (grades F2 to 4), while 2 had F0 and 11 had F1 fibrosis. Only 4 out of 11 KTRs without RBV treatment were investigated, all with mild liver fibrosis (F1).

### 3.3. Risk Factors for RBV Treatment Failure

Six patients received treatment for less than three months, 14 for approximately three months, eight for 3–6 months, six for approximately six months and 14 for longer than six months. The mean time to viral rebound was 4.0 ± 2.9 months. To analyze the risk factors for insufficient viral clearance after the RBV treatment, we subdivided all RBV-treated KTRs according to the treatment outcome (Table 4).

Treatment success was defined as viral clearance and an SVR after the first treatment round with RBV (n = 29). Treatment failure was defined as ongoing viremia or a viral rebound after an initial viral clearance (n = 19). KTRs with treatment failure had a longer mean treatment duration with RBV compared with the KTRs with treatment success (471 ± 511.5 vs. 118 ± 58.5 days, *p* < 0.01). The mean RBV doses in both groups did not differ, but we asked whether the start dose was inadequately low or whether the RBV-induced side effects, especially anemia, led to an interruption of the RBV treatment, dose reduction and, as a consequence, higher rates of viral relapse. Detailed characteristics are depicted in Table 5. We observed a lower median RBV dose at the initiation of treatment in patients with RBV treatment failure vs. patients with successful viral clearance (300 vs. 400 mg). With respect to the high rates of relevant anemia in both cohorts (78.9 and 58.6%), the rates of EPO treatment and the start of EPO treatment during RBV treatment (73.6 and 55.1%) were not significantly higher in patients with RBV treatment failure, but there was a significant delay of EPO initiation in patients with treatment failure (after 67.6 ± 49.5 vs. 16.0 ± 15.2 days).

Only 4 out of 19 (21%) patients with ongoing viremia or a viral rebound reached an SVR in the long-term after a median follow-up time of 5 (range 2.0–5.0) years since the diagnosis of HEV viremia. A negative HEV PCR by week four was observed in 69% of KTRs with treatment success vs. in 47.4% of KTRs without (*p* = 0.089), which may be an independent predictive factor for a positive treatment outcome.

Although not significant, we observed a trend toward older mean age in KTRs with treatment failure (56.1 ± 13.7 years) vs. KTRs with viral clearance (49.5 ± 15.7 years) (*p* = 0.054, Figure 1A). Univariable analysis revealed age > 60 years as a risk factor for RBV treatment failure with an odds ratio of 7.50 (95% CI 1.20–47.05). Besides this, the KTRs with treatment failure had a significantly lower mean body mass index (BMI) compared with the KTRs with viral clearance (22.2 ± 4.2 kg/m^2^ vs. 25.3 ± 4.7 kg/m^2^, *p* < 0.05, Figure 1B).

Moreover, a BMI ≤ 20 kg/m^2^ was significantly associated with failure to achieve viral clearance (10.3% in the group with treatment success vs. 42.1% in the group with treatment failure, *p* < 0.05, Table 4). On further multivariable analysis, only age > 60 years and a BMI ≤ 20 kg/m^2^ were significantly associated with RBV treatment failure, with odds ratios of 4.34 (95% CI 1.05–17.96) and 7.73 (95% CI 1.53–39.05), respectively (Table 6).

Furthermore, albeit not significant, we observed a higher rate of deterioration in kidney function and proteinuria during hepatitis E viremia (decrease of eGFR ≥ 5 mL/min) in the KTRs with treatment failure vs. treatment success (78.9% vs. 51.7%, *p* = 0.07). To consider RBV mutations as a reason for treatment failure, a mutation analysis in the HEV polymerase was performed in 12 out of 19 patients with RBV treatment failure; the results were found to be positive in 7 patients.

### 3.4. Eating Habits of KTRs with or without HEV Infection

Finally, we utilized a questionnaire to evaluate the risk behavior with respect to eating habits by referring to the time before disease onset or diagnosis (Table 7).

Dietary surveys were available from 40 out of 59 patients with HEV infection. To compare eating habits with a control group of KTRs, the same questionnaire was answered by 80 control KTRs (to reach a 1:2 ratio) with no evidence of HEV infection during the same observation period. These patients were mainly not part of the original control group since, due to the retrospective nature of the analysis, not enough dietary information was available from KTRs without HEV infection. There was no KTR in close contact with house pigs in both groups and there were no professional hunters amongst the KTRs. As expected, most KTRs in our cohort ate meat. There were only three vegetarians amongst all patients who filled in the questionnaire. Remarkably, despite education and advice regarding the avoidance of uncooked or undercooked meat, approximately one out of five KTRs consumed raw minced pork and beef meat, and 86.9% of the HEV group patients ate raw sausage (such as salami) and 78.9% ate ham (compared with 70.5% and 59%, respectively, in the CG). A significantly higher number of KTRs with HEV infection consumed spreadable raw sausages (such as German “Teewurst”), cured pork meat, boiled sausages and blood sausage (Table 7). Regarding the preparation of meat at home, both meat processing at home and touching raw meat by bare hands were more common in KTRs with HEV infection.

In contrast, there was a trend toward less meat consumption and more consumption of fish, raw vegetables and raw milk (mostly cheese products) in the CG.

### 3.5. LongTerm Follow-Up and Allograft Outcome

The median time of the follow-up in the HEV cohort was 4.2 (IQR 2.2, 5.8) years since the diagnosis of HEV infection and 3.8 (IQR 1.8, 5.6) years in the CG. During four years of patient individual follow-up, graft failures occurred in 8 (13.6%) patients of the HEV cohort, with a median time to loss of 1.4 (IQR 0.5, 2.4) years, and in 16 (6.4%) patients of the CG, with a median time to loss of 1.6 (IQR 0.7, 2.5) years.

The survival of the KTRs with hepatitis E was 89.8% (6 out of 59 died, two of those with a functioning graft) vs. 89.6% (26 out of 251 died, four of those with a functioning graft) in the CG. Thus, deathcensored graft survival was 89.8 vs. 95.2%. Rates of deathcensored graft losses and deaths did not differ significantly between both groups, as described by the Kaplan–Meier estimator shown in Figure 2A,B (*p* = 0.160 vs. *p* = 0.869).

## 4. Discussion

We found that intensive immunosuppressive treatments and the consumption and/or processing of undercooked pork meat were associated with hepatitis E infection in patients after SOT. In addition, higher age and lower BMI were risk factors for persistent hepatitis E viremia and RBV treatment failure, resulting in lower rates of remission and sustained viral responses in the long term.

The risk of infections after kidney transplantation is influenced by the intensity of immunosuppression, e.g., used for induction therapy, maintenance therapy and antirejection treatments. Our findings are in line with Kamar et al. [5] and our previous evaluation [4], which showed deleterious effects of previous T or Bcell targeting therapies regarding the incidence of hepatitis E infection and the risk for viral persistence in kidney transplant recipients, even many years after transplantation. This is also in line with the observation made by Kamar et al., who found that tacrolimus levels and steroid doses were lower in KTRs with viral clearance of hepatitis E viremia compared with KTRs with persistent viremia [27]. A small case series of five patients with immunocompromised conditions and previous treatments containing rituximab (RTX) demonstrated a longterm risk for HEV infection with persistent viremia or relapses despite treatment with RBV, where only one out of five reached an SVR after 6 months of treatment [28]. Another case series described chronic HEV infections after the use of RTX and bendamustine in lymphoma patients [29]. Thus, intense immunosuppressive treatment regimens beyond standard induction, such as the use of basiliximab, and standard maintenance therapy carry the risk for higher rates of HEV infection. In our cohort, a history of antirejection treatment with highdose steroids, thymoglobuline, and/or RTX due to cellular or antibodymediated rejections (with the latter often including a combination of steroids, plasmapheresis, rituximab and IVIG) in the past were associated with a higher incidence of HEV infection. Consequently, in terms of the risk of HEV infections, “blind” rejection treatments, e.g., with highdose steroids after SOT, should be avoided, and patienttailored treatments for antibodymediated rejection (ABMR) are warranted. Along the same line, the use of RTX should be considered carefully.

While maintenance immunosuppression in all KTRs was dominated by the intake of triple immunosuppression of CNI, MPA and lowdose CS, a trend toward a more frequent use of the CNI tacrolimus compared with cyclosporine A was found in both patients with persistent viremia and patients unable to clear the virus without RBV treatment. Moreover, the use of CyA instead of tacrolimus was significantly higher in our control cohort of KTRs without HEV infection and correlated significantly with a potentially lower risk for HEV infection. This is possibly related to the fact that CyA was the first available CNI inhibitor in kidney transplantation. It was used for maintenance therapy for many years and kept in case of stable allograft function. Second, a switch from tacrolimus to CyA was used as an alternative option in case of tacrolimusinduced side effects or recurrent episodes of infection. We believe that the intensity of immunosuppression matters and that tacrolimus instead of CyA might predispose patients to more viral infections, including HEV infection.

We have to emphasize that we did not screen all KTRs from our center for HEV but only patients with (unexplained) elevated liver enzymes. Because of this, we think that by and large, the HEV seroprevalence in our cohort of KTRs (43.9%) was higher compared with other European studies (approximately 10–20% [30,31,32], with the exception of several French cohorts with a higher seroprevalence in general [33,34,35]). Serology for IgG and IgM may persist even years after infection, which underlines the importance of PCR tests to differentiate between previous and acute infection. The prevalence of KTR with RTPCRconfirmed hepatitis E was 19% in our cohort, which was substantially higher compared with other studies, with a range from 0.9% to 3.5% [30,36,37]. The reason why is likely related to our clinical practice to test KTRs only in the case of elevated liver enzymes, which highlights the importance to screen for HEV in this special cohort.

We observed a potential HEVassociated kidney involvement with a rise in proteinuria in KTR with persistent HEV viremia and RBV treatment compared with KTRs with remission without RBV treatment. This indicates that an increase in proteinuria and deterioration in kidney function during HEV infection is not only related to chronic impairment of allograft function but also a direct result of HEV infection. Certain findings in kidney transplant histology, as described in our earlier paper, increase the likelihood of HEVinduced GN and influence whether it resolves after viral clearance [4]. This is in line with previous reports, demonstrating kidney involvement with or without a loss of eGFR or a rise in proteinuria [38,39]. HEVmediated kidney damage may be a result of a direct or indirect cytopathic injury caused by the virus itself or by immunecomplexmediated mechanisms, as demonstrated in vitro studies [40,41]. Similar to hepatitis B and C, HEVassociated cryoglobulinemic vasculitis was described [42,43], but we did not observe this in our patient cohort.

The number of patients without a need for further RBV treatment was low (11 vs. 48). In this small cohort, we could not identify significant predictors for spontaneous remission vs. continuous viremia. The more frequent use of Tac (instead of CyA) points toward more intense maintenance IS as a potential risk factor for the failure of viral clearance.

Chronic hepatitis E can lead to liver fibrosis and cirrhosis [3]. Noninvasive methods, such as transient elastography or shear wave elastography, have received substantial attention due to their easy access, reproducibility and accuracy in common liver diseases, such as nonalcoholic fatty liver disease and hepatitis B and C [23,44]. From our cohort of 59 KTRs with HEV infection, transient elastography or shear wave elastography measurements were available from 32 KTRs and demonstrated substantial liver stiffness in 15. Since this was a retrospective study, we could not include control KTRs without hepatitis E, but this was shown in a previous study by Schulz et al., where liver stiffness was significantly higher in KTRs with a history of hepatitis E than in control KTRs [21]. The use of serial measurements for liver surveillance and changes in liver elasticity during HEV disease or after treatment should be investigated in future prospective studies.

The high rate of KTRs with a recurrence of HEV replication after the end of or during RBV treatment (19 out of 48) warrants further investigations regarding risk factors. One explanation might be an inadequately low RBV dose at the initiation of treatment, as observed in our cohort in about 50% of cases. The high frequency of RBVinduced anemia might have triggered the decision to start with a lower RBV dose. In addition, the significant delay of EPO therapy in patients with RBV treatment failure likely contributed to a higher number of relevant anemia, RBV dose reduction and, as a consequence, more relapses or an incomplete viral clearance. We emphasize the need for adequate dosing of RBV to treat hepatitis E infection and we recommend immediate or very early substitution of EPO in the case of anemia or a drop in hemoglobin. The rebound or reinfection rate in our study was higher than reported by Kamar et al. [8,9], where the authors used a higher initial median dose of 600 mg per day in a retrospective study with an HEV rebound in 10 out of 59 patients after SOT with 37 KTRs and 5 combined kidney–pancreas transplant recipients [9]. Another reason to detect a higher rate of relapses might be the longer observation and follow-up period of our study. Resistance to RBV due to mutations is another major concern. Mutations in the HEV polymerase were found in approximately 50% of our KTRs with treatment failure. Data regarding the frequency and impact of mutations leading to RBV resistance are scarce but should be considered in patients with a difficulttotreat HEV infection [20,45,46]. The impact on treatment outcomes has to be investigated in larger studies.

In our cohort, a lower BMI, especially ≤20 kg/m^2^, was a risk predictor in the KTRs with failure to sustain viral clearance. A low BMI might indicate malnutrition or cachexia with a more enhanced immunocompromised condition. Nutritional advice during patient care and avoidance of weight loss with the detection of factors associated with cachexia, e.g., undetected inflammation, are important. Along the same line, older age (>60 years) tended to be a risk factor for an unfavorable outcome [47,48]. Moreover, an early viral response to RBV may have an impact on the overall treatment response. Although not significant, we observed a higher rate of negative HEV PCRs by week four in KTRs with viral clearance after RBV treatment and a subsequent SVR compared with KTRs with relapsing courses or failure to achieve an SVR. Kamar et al. demonstrated that an initial decrease in the HEV viral load ≥ 0.5 log copies/mL had an 88% positive predictive value for an SVR [49]. Along the same line, a negative viral load one month after treatment was associated with a higher sustained virologic response [9,34]. However, the predictive value of this measure needs to be validated in larger studies. Overall, we suggest an RBV treatment time of at least three months, with a prolonged course of 6–9 months or even longer than 9 months in the case of persistent viremia or relapsing disease.

Longterm allograft and patient outcomes did not differ between KTRs with hepatitis E and control patients. In a Scottish multicenter retrospective study, 511 cases with an HEV infection were reported between 2013 to 2018, with a mortality rate of 3.3% and chronic courses of HEV infection were a positive predictor for mortality in their study [50]. However, we did not observe an increased hepatitisErelated mortality in our study.

Finally, after organ transplantation, we educate patients to avoid undercooked meat, as well as potentially contaminated food products. Regarding meat, a cooking temperature of 71 °C for 20 min is required to fully inactivate the virus, but especially with readytoeat products, there is a lack of detailed knowledge regarding manufacturing processes. Moreover, drying procedures, such as for raw ham, do not guarantee virus inactivation. Indeed, HEV RNA could be detected in commercial pork sausages, pork liver and liver pâté in German groceries [51,52,53].

We assume that patients are either not aware of or underestimate the risk of HEV acquisition by potentially undercooked products, such as raw sausages (in Germany typically through “Salami”, “Mettwurst” and “Leberwurst”), boiled sausages (e.g., all kinds of “Brühaufschnitt”) and “Wiener” sausage. Interestingly, there is a male predominance regarding the consumption of undercooked meat products among KTRs. Given HEV infections even many years after SOT, we assume a general loss of vigilance in the long term.

We suggest continuous education of KTRs regarding food habits and emphasize a detailed anamnesis after diagnosis of HEV infection to find relevant sources for transmission and possibly persistent viremia. Moreover, we recommend that meat processing at home with bare hands should be avoided, e.g., by using gloves, and careful cleaning of hands and knives with soap is advised afterward. Since KTRs in the control cohort ate products such as seafood, raw vegetables and salads more frequently than KTRs in the hepatitis E cohort, these products are probably not the main cause of HEV transmission.

A limitation of the study was its retrospective nature with a singlecenter cohort. Laboratory assessment for HEV RNA and serology was only performed for KTRs with elevated liver enzymes.

## 5. Conclusions

The treatment of HEV infection remains challenging and intense immunosuppression is a risk factor for HEV infection with potentially chronic courses and deleterious outcomes. Food that contains undercooked pork meat is a relevant health issue, for which intensified patient education appears warranted, especially in patients with intense immunosuppressive regimens, and even many years after SOT.

## Figures and Tables

**Figure 1 pathogens-12-00850-f001:**
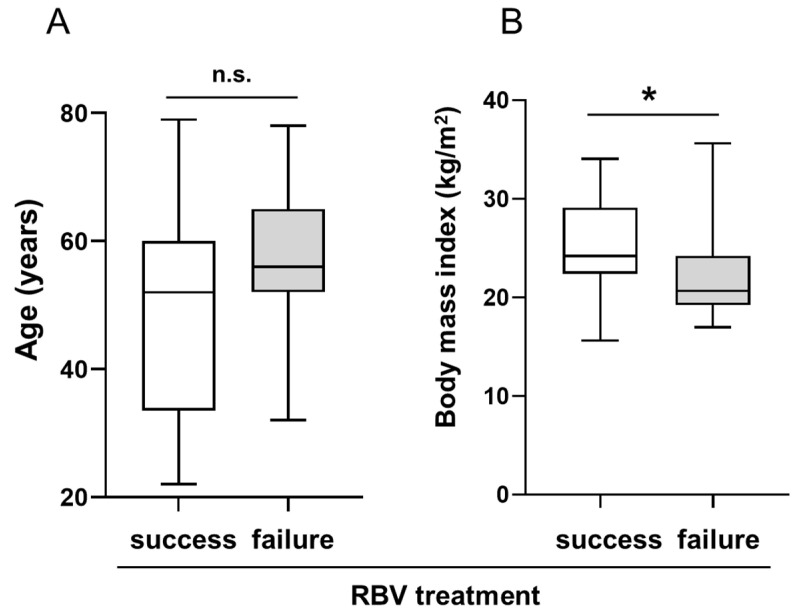
Impacts of age and body mass index (BMI) of ribavirin (RBV) treatment outcome in kidney transplant recipients (KTRs) with HEV infection. (**A**) KTRs with higher age showed a non-significant trend toward more frequent RBV treatment failure vs. treatment success (*p* = 0.054). (**B**) A lower BMI correlated significantly with a higher rate of RBV treatment failure. n.s.—not significant, * *p* < 0.05.

**Figure 2 pathogens-12-00850-f002:**
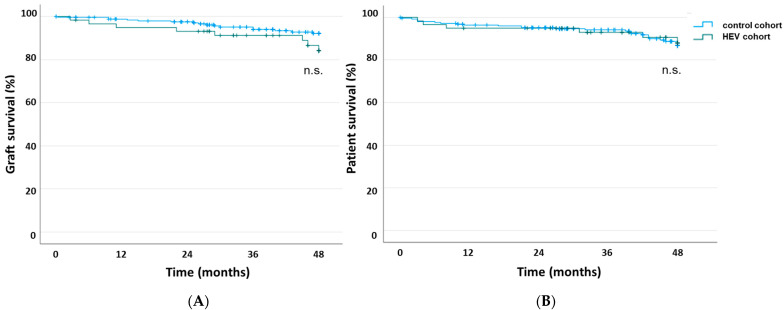
Kaplan–Meier estimates of allograft survival (**A**) and patient survival (**B**) in KTRs with HEV infection (green line) vs. KTRs without HEV infection (blue line) after diagnosis of hepatitis E or after diagnostic testing (control cohort). n.s.—not significant.

**Table 1 pathogens-12-00850-t001:** Baseline characteristics of KTRs with elevated liver enzymes by status for HEV infection.

Patient Characteristics	KTRs with HEV Infection	KTRs w/o HEV Infection	Statistical Group	Univariable OR (95% CI)
	N = 59	N = 251	Difference,	
	N (%)	N (%)	*p*-Value	
Gender				
Male	46 (78.0)	141 (56.2)	0.002	2.76 (1.42–5.36)
Female	13 (22.0)	110 (43.8)		
Age				
Mean (y)	50.9 ± 15	54.5 ± 13.3	0.103	
<40	16 (27.1)	41 (16.3)		Reference
40–60	27 (45.8)	127 (50.6)	0.063	1.91 (0.98–3.70)
>60	16 (27.1)	83 (33.1)	0.439	0.75 (0.40–1.42)
Renal disease				
Glomerulonephritis	20 (33.9)	66 (26.3)	0.260	1.44 (0.78–2.6)
Cystic kidney disease	7 (11.9)	44 (17.5)	0.335	0.63 (0.27–1.49)
Diabetic nephropathy	7 (11.9)	33 (13.1)	1.000	0.89 (0.37–2.12)
Hypoplastic kidneys	4 (6.8)	13 (5.2)	0.541	1.33 (0.42–4.24)
Other	6 (10.2)	44 (17.5)	0.237	0.53 (0.22–1.32)
Unknown	16 (27.1)	44 (17.5)	0.101	1.75 (0.91–3.39)
Co-morbidities				
Hypertension	47 (79.7)	201 (80.0)	1.000	0.97 (0.48–1.97)
Diabetes mellitus	14 (23.7)	69 (27.5)	0.626	0.82 (0.42–1.60)
Arteriosclerosis	19 (32.2)	64 (25.5)	0.328	1.39 (0.75–2.57)
History of cancer	12 (20.3)	34 (13.5)	0.221	1.63 (0.79–3.38)
Chronic liver disease	8 (13.6)	52 (20.7)	0.272	0.60 (0.27–1.34)
BMI (kg/m^2^)	24.2 ± 4.7	25.5 ± 4.9	0.114	
Time after transplantation (m)	93 ± 80	92 ± 85	0.631	
Type of transplant			0.688	
Kidney	51 (86.4)	226 (90.0)		
Pancreas–kidney	5 (8.5)	17 (6.8)		
Other multi-visceral	3 (5.1)	8 (3.2)		
Type of donation			0.109	
Postmortem	41 (69.5)	186 (74.1)		
Living donation	12 (20.3)	54 (21.5)		
ABOi living donation	6 (10.2)	9 (3.6)		
Re-transplant	10 (16.9)	27 (10.8)	0.187	1.69 (0.77–3.73)
Immunosuppressive regimen				
Tacrolimus	49 (83.0)	179 (71.0)	0.072	1.97 (0.95–4.10)
Cyclosporine A	3 (5.1)	52 (20.7)	0.004	0.21 (0.06–0.68)
Mycophenolic acid	57 (96.6)	222 (88.4)	0.088	3.72 (0.86–16.07)
mTOR inhibitor	6 (10.2)	10 (4.0)	0.093	2.73 (0.95–7.84)
Steroid	47 (79.7)	196 (78.1)	0.862	1.10 (0.55–2.22)
Belatacept	3 (5.1)	11 (4.4)	0.735	1.17 (0.32–4.33)
Triple IS	45 (76.3)	176 (70.1)	0.425	1.37 (0.71–2.65)
Dual IS	14 (23.7)	75 (30.0)	0.424	0.72 (0.38–1.40)
Previous treatment of any rejection	29 (49.2)	87 (34.7)	0.051	1.82 (1.03–3.23)
Use of thymoglobulin	20 (33.9)	59 (23.5)	0.134	1.67 (0.90–3.08)
Use of high dose steroids	33 (55.9)	83 (33.1)	0.002	2.57 (1.44–4.58)
Use of rituximab	10 (16.9)	14 (5.6)	0.007	3.46 (1.45–8.23)
Treatment of aABMR	7 (11.9)	6 (2.4)	0.004	5.47 (1.77–18.96)
Baseline eGFR	52 ± 21	51 ± 22		
eGFR at diagnosis (mL/min)	52 ± 20	52 ± 22		
Creatinine at diagnosis (mg/dL)	1.58 ± 0.6	1.64 ± 0.9	0.856	
Baseline proteinuria (mg/g creatinine) *	344 ± 551	302 ± 649	0.667	
Peak proteinuria at liver enzyme elevation (mg/g creatinine)	713 ± 919	580 ± 1045	0.671	
Rise in proteinuria during observation period	15/50 (30.0)	31/215 (14.4)	0.009	
Liver enzymes				
ALT max (U/L, ref < 35)	224 ± 230	118 ± 126	<0.001	
AST max (U/L, ref < 31)	135 ± 135	83 ± 82	<0.001	
gGT max (U/L, ref 5–36)	222 ± 175	275 ± 381	0.948	
Serology			<0.001	
IgG pos	56/56 (100)	80 (31.9)		
IgM pos	49/56 (87.5)	11 (4.4)		
IgM and IgG positive	49/56 (87.5)	11 (4.4)		
IgM and IgG negative	0 (0)	171 (68.1)		
Not done	3 (5.1)	0 (0)		

Data were expressed as mean (±standard deviation) or numbers (n) unless stated otherwise. Ref—reference group. * control: at the time of diagnostic testing. KTR—kidney transplant recipient, HEV—hepatitis E virus, y—years, m—months, BMI—body mass index, ABOi—ABO incompatible, mTOR—mammalian target of rapamycin, IS—immunosuppression, h/o—history of, aABMR—active antibody-mediated rejection, eGFR—estimated glomerular filtration rate, ALT—alanine aminotransferase, AST—aspartate transferase, gGT—gamma-glutamyl transferase, Ig—immunoglobulin.

**Table 2 pathogens-12-00850-t002:** Predictors of HEV infection after SOT identified in the multivariable analysis.

Variable	Odds Ratio (95% CI)	*p*-Value
Male sex	3.19 (1.58–6.43)	0.001
Previous use of high-dose steroids	2.13 (1.15–3.94)	0.016
Previous use of rituximab	2.96 (1.13−7.75)	0.027
Use of CyA as maintenance IS	0.24 (0.07–0.80)	0.021

**Table 3 pathogens-12-00850-t003:** Patient characteristics (persistent HEV viremia vs. remission without RBV treatment).

Patient Characteristics	Persistent HEV Viremia (RBV-Treated)	Remission Without RBV Treatment	Statistical Group Difference, *p*-Value	Univariable OR (95% CI)
	N = 48	N = 11		
	N (%)	N (%)		
Gender			**0.426**	**0.30 (0.04–2.59)**
Male	36 (75)	10 (90.9)		
Female	12 (25)	1 (9.1)		
Age				
Mean (y)	52 ± 15	46 ± 14	0.134	
<40	12 (25)	4 (36.4)		Reference
40–60	21 (43.75)	6 (54.5)	1.000	0.86 (0.20–3.66)
>60	15 (31.25)	1 (9.1)	0.333	5.00 (0.49–50.8)
Renal disease				
Glomerulonephritis	17 (29.2)	3 (27.3)	0.734	1.46 (0.34–6.25)
Cystic kidney disease	7 (14.6)	0 (0)	0.328	1.17 (1.04–1.32)
Diabetic nephropathy	5 (10.4)	2 (18.2)	0.604	0.52 (0.09–3.14)
Hypoplastic kidneys	4 (8.3)	0 (0)	1.000	1.09 (1.00–1.19)
Other	4 (8.3)	2 (18.2)	0.310	0.41 (0.07–2.58)
Unknown	12 (25)	4 (36.4)	0.468	0.58 (0.15–2.35)
Co-morbidities				
Hypertension	37 (77.1)	10 (90.9)	0.431	0.34 (0.39–2.93)
Diabetes mellitus	10 (20.8)	4 (36.4)	0.432	0.46 (0.12–1.89)
Arteriosclerosis	16 (33.3)	3 (27.3)	1.000	1.33 (0.31–5.72)
History of cancer	11 (22.9)	1 (9.1)	0.431	2.97 (0.34–25.9)
Chronic liver disease	7 (14.6)	1 (9.1)	1.000	1.71 (0.19–15.5)
BMI (kg/m^2^)	24 ± 4.7	25 ± 5.0	0.350	
Time after transplantation (m)	96 ± 83	80 ± 67	0.647	
Type of transplant			0.329	
Kidney	42 (87.5)	9 (81.2)		
Pancreas–kidney	3 (6.25)	2 (18.2)		
Other multi-visceral	3 (6.25)	0 (0)		
Type of donation			0.424	
Postmortem	33 (68.75)	8 (72.3)		
ABOc living donation	11 (22.9)	1 (9.1)		
ABOi living donation	4 (8.3)	2 (18.2)		
Re-transplantation	7 (14.6)	3 (27.3)	0.376	0.46 (0.10–2.15)
Immunosuppressive regimen				
Tacrolimus	41 (85.4)	8 (72.7)	0.376	2.20 (0.47–2.15)
Cyclosporine	1 (2.1)	2 (18.2)	0.086	0.10 (0.08–1.17)
Mycophenolic acid	47 (97.9)	10 (90.9)	0.341	4.70 (0.27–81.6)
mTOR inhibitor	6 (12.5)	0 (0)	0.582	1.14 (1.03–1.27)
Steroid	40 (83.3)	7 (63.6)	0.209	2.86 (0.67–12.1)
Belatacept	2 (4.2)	1 (9.1)	0.468	0.44 (0.04–5.28)
Triple IS	39 (81.3)	6 (54.5)	0.110	3.61 (0.90–14.5)
Dual IS	9 (18.8)	5 (45.5)	0.110	0.28 (0.07–1.11)
Reduction/switch of IS after diagnosis	16 (33.3)	3 (27.3)	1.000	1.33 (0.31–5.72)
Previous treatment of any rejection	26 (54.2)	3 (27.3)	0.181	3.15 (0.74–13.4)
Use of thymoglobulin	16 (33.3)	4 (36.4)	1.000	0.88 (0.22–4.49)
Use of high-dose steroids	29 (60.4)	4 (36.4)	0.187	2.67 (0.69–10.4)
Use of rituximab	9 (18.8)	1 (9.1)	0.670	2.31 (0.26–20.4)
Treatment of aABMR	7 (14.6)	0 (0)	0.328	1.17 (1.04–1.32)
eGFR at diagnosis (mL/min)	47 ± 30	48 ± 56	0.325	
Creatinine at diagnosis (mg/dL)	1.6 ± 0.6	1.5 ± 0.6	0.546	
eGFR during HEV infection				
Stable	18 (37.5)	8 (72.7)		
Deterioration (≥5 mL/min)	30 (62.5)	3 (27.3)	0.046	
Baseline proteinuria (mg/g creatinine)	361 ± 578	72 ± 66	0.014	
Peak proteinuria during HEV infection (mg/g creatinine)	735 ± 959	164 ± 376	0.008	
Rise in proteinuria during HEV viremia (≥300 mg/g crea)	15/42 (35.7)	0/8 (0)	0.086	
Liver enzymes				
ALT max (U/L, ref < 35)	226 ± 249	217 ± 120	0.414	
AST max (U/L, ref < 31)	133 ± 138	140 ± 130	0.869	
gGT max (U/L, ref 5–36)	233 ± 183	169 ± 126	0.217	
Max HEV RNA (mio cop/mL)	1.8 ± 2.1	0.7 ± 1.5	0.010	
Grade of liver fibrosis				
0	2 (4.2)	0		
1	11 (22.9)	4 (36.4)		
2	3 (6.25)	0		
3	9 (18.75)	0		
4	3 (6.25)	0		
Not done	20 (41.7)	7 (63.4)		
Outcome				
Time of viremia (d)	278 ± 203	126 ± 104	0.002	
(range: min–max)	(84–989)	(21–334)		
Relapse	17 (29.2)	0 (0)	0.040	1.59 (1.27–1.98)
SVR (>6 m)	33 (68.75)	11 (100)	0.050	

Data were expressed as mean (±standard deviation) or numbers (n) unless stated otherwise. KTR—kidney transplant recipient, HEV—hepatitis E virus, y—years, m—months, BMI—body mass index, ABOc—ABO compatible, ABOi—ABO incompatible, mTOR—mammalian target of rapamycin, IS—immunosuppression, h/o—history of, aABMR—active antibody-mediated rejection, eGFR—estimated glomerular filtration rate, ALT—alanine aminotransferase, AST—aspartate transferase, gGT—gamma-glutamyl transferase.

**Table 4 pathogens-12-00850-t004:** Characteristics of RBV-treated KTRs by treatment outcome.

Patient Characteristics	RBV Treatment Failure *	RBV Treatment Success	Statistical Group Difference, *p*-Value	Univariable OR (95% CI)
	N = 19	N = 29		
	N (%)	N (%)		
Gender				
Male	15 (78.9)	21 (72.4)	0.739	0.70 (0.18–2.76)
Female	4 (21.1)	8 (27.6)		
Age				
Mean (y)	56.1 ± 13.7	49.5 ± 15.7	0.054	
<40	2 (10.5)	10 (34.5)		Ref
40–60	8 (42.1)	13 (44.8)	0.259	3.08 (0.53–17.80)
>60	9 (47.4)	6 (20.7)	0.047	7.50 (1.20–47.05)
Co-morbidities				
Hypertension	16 (84.2)	21 (72.4)	0.488	2.03 (0.46–8.90)
Diabetes mellitus	5 (26.3)	5 (17.2)	0.487	1.71 (0.42–6.98)
Arteriosclerosis	9 (47.9)	7 (24.1)	0.124	2.83 (0.82–9.76)
History of cancer	5 (26.3)	6 (20.7)	0.732	1.40 (0.35–5.34)
Chronic liver disease	5 (26.3)	2 (6.9)	0.097	4.82 (0.83–28.1)
BMI (kg/m^2^)	22.2 ± 4.2	25.3 ± 4.7	0.019	
BMI ≤ 20 kg/m^2^	8 (42.1)	3 (10.3)	0.032	5.8 (1.29–26.25)
Time after transplantation (m)	90 ± 83	99 ± 85	0.524	
Type of transplant				
Kidney	16 (84.2)	26 (89.7)		Ref
Pancreas–kidney	1 (5.3)	2 (6.9)	1.000	0.81 (0.07–9.70)
Other multi-visceral	2 (10.5)	1 (3.4)	0.555	3.25 (0.27–38.81)
Immunosuppressive regimen				
Tacrolimus	16 (84.2)	25 (86.2)	1.000	0.85 (0.17–4.33)
Cyclosporine A	0 (0)	1 (3.4)	1.000	0.60 (0.47–0.75)
Mycophenolic acid	19 (100)	28 (96.6)	1.000	1.68 (1.33–2.12)
mTOR inhibitor	2 (10.5)	4 (14.0)	1.000	0.74 (0.12–4.47)
Steroids	17 (89.5)	23 (79.3)	0.451	2.22 (0.40–12.4)
Belatacept	2 (10.5)	0 (0)	0.152	2.71 (0.19–3.95)
Triple IS	16 (84.2)	23 (79.3)	1.000	1.39 (0.30–6.40)
Dual IS	3 (15.8)	6 (20.7)	1.000	0.72 (0.16–3.31)
Previous treatment of any rejection	8 (42.1)	18 (62.1)	0.239	2.25 (0.69–7.32)
Use of thymoglobulin	6 (31.6)	10 (34.5)	1.000	1.14 (0.33–3.92)
Use of high-dose steroids	11 (57.9)	18 (62.1)	1.000	1.19 (0.37–3.87)
Use of rituximab	4 (21.1)	5 (17.2)	1.000	0.78 (0.18–3.38)
Treatment of aABMR	2 (10.5)	5 (17.2)	0.687	1.77 (0.31–10.2)
Renal function				
Creatinine at diagnosis (mg/dL)	1.6 ± 0.6	1.6 ± 0.6	0.847	
eGFR at diagnosis (mL/min)	49 ± 18	45 ± 36	0.819	
Minimal eGFR at disease (mL/min)	35 ± 20	45 ± 20	0.362	
Renal function during HEV viremia				
Stable	4 (21.1)	14 (48.3)		Ref
Deterioration (≥5 mL/min)	15 (78.9)	15 (51.7)	0.073	3.5 (0.93–13.13)
Baseline proteinuria (mg/g crea)	253 ± 555	432 ± 591	0.292	
Peak proteinuria during HEV (mg/g crea)	1004 ± 1243	560 ± 684	0.167	
Rise in proteinuria during HEV (≥300 mg/g crea)	8/16 (50)	7/26 (26.9)	0.188	2.71 (0.73–10.04)
Liver enzymes				
ALT max (U/L, ref < 35)	215 ± 256	232 ± 249	0.829	
AST max (U/L, ref < 31)	157 ± 105	138 ± 127	0.927	
gGT max (U/L, ref 5–36)	248 ± 184	224 ± 185	0.699	
HEV RNA max (mio cop/mL)	1.8 ± 2.3	1.8 ± 1.9	0.558	
Duration of RBV treatment (d)	471 ± 511.5	118 ± 58.5	<0.001	
(range: min–max)	(45–2155)	(39–310)		
RBV dosage (mg/d)	363 ± 201	381 ± 152	0.332	
Hemolytic anemia	15 (78.9)	17 (58.6)	0.111	3.53 (0.83–14.9)
Side effects of treatment	18 (95)	23 (79)	0.219	4.70 (0.52–42.6)
Negative HEV PCR 4 weeks after start of treatment	9 (47)	20 (69)	0.120	0.32 (0.09–1.10)
Remission after last treatment	13 (68)	29 (100)	0.002	
SVR	4 (21)	29 (100)	<0.001	

Data are expressed as mean (±standard deviation) unless stated otherwise. * RBV treatment failure was defined as ongoing viremia despite RBV intake or viral rebound after initial viral clearance. y—years, m—months, BMI—body mass index, eGFR—estimated glomerular filtration rate, crea—creatinine, mTOR—mammalian target of rapamycin, IS—immunosuppression, aABMR—active antibody-mediated rejection, HEV—hepatitis E virus, RBV—ribavirin, PCR—polymerase chain reaction, SVR—sustained virologic response.

**Table 5 pathogens-12-00850-t005:** Ribavirin dosing, ribavirin-induced anemia and rate of erythropoietin treatment according to status for treatment outcome.

Characteristics	RBV Treatment Failure	RBV Treatment Success	Statistical Group Difference, *p*-Value
	N = 19	N = 29	
	N (%)	N (%)	
Duration of RBV treatment (d)	471 ± 511.5	118 ± 58.5	<0.001
(range: min–max)	(45–2155)	(39–310)	
Mean RBV dosage (mg/d)	363 ± 201	381 ± 152	0.332
Median start dose	300	400	0.347
(range: min–max)	(200–800)	(200–1000)	
Start dose lower than GFR-adapted dosing	9 (47.4)	13 (44.8)	1.000
Increase of dosage	6 (31.6)	1 (3.4)	0.770
- After a mean time of (d)	204 ± 194	14	0.500
Reduction of dosage	6 (31.6)	7 (24.1)	0.741
- After a mean time of (d)	71.5 ± 30.8	63.4 ± 43.9	0.366
RBV-induced anemia	15 (78.9)	17 (58.6)	0.111
Interruption due to hemolytic anemia	3 (15.8)	3 (10.3)	0.662
- After a mean time of (d)	113.3 ± 61.1	67.7 ± 23.7	0.400
EPO treatment before RBV	7 (36.8)	11 (37.9)	1.000
EPO start after RBV	7 (36.8)	5 (17.2)	0.168
- After a mean time of (d)	67.6 ± 49.5	16.0 ± 15.2	0.048
Increase in EPO dosage	12 (63.2)	14 (48.3)	0.245

Data are expressed as mean (±standard deviation) unless stated otherwise. RBV—ribavirin, d—days, GFR—glomerular filtration rate, EPO—erythropoietin.

**Table 6 pathogens-12-00850-t006:** Predictors of KTRs with RBV treatment failure identified in multivariable analysis.

Variable	Odds Ratio (95% CI)	*p*-Value
Age > 60 years	4.34 (1.05–17.96)	0.043
BMI ≤ 20 kg/m^2^	7.73 (1.53–39.05)	0.013

KTR—kidney transplant recipient, RBV—ribavirin, BMI—body mass index.

**Table 7 pathogens-12-00850-t007:** Food habits of KTRs with elevated liver enzymes by HEV infection status.

Patient Characteristics	KTRs with HEV Infection N = 40 N (%)	KTRs w/o HEV Infection N = 80 N (%)	Statistical Group Difference, *p*-Value	Univariable OR (95% CI)
**Gender**				
Male	30 (75)	51 (63.75)		
Female	10 (25)	29 (36.25)		
**Age**				
Mean (y)	52 ± 13.6	55.6 ± 13.6	0.241	
**Fish**	(39)	(79)		
At least weekly	6 (15.4)	39 (49.4)	0.003	0.11 (0.03–0.46)
Less than weekly	27 (69.2)	36 (45.6)	0.359	0.54 (0.15–1.87)
Never	6 (15.4)	4 (5.1)		Ref
**Seafood**	(40)	(76)		
At least weekly	0 (0)	1 (1.3)	1.000	0.60 (0.54–0.73)
Less than weekly	3 (7.5)	18 (25)	0.043	0.26 (0.07–0.95)
Never	36 (90)	56 (73.7)		Ref
**Salad**				
At least weekly	28 (70)	67 (83.75)	0.511	0.42 (0.03–6.92)
Less than weekly	11 (27.5)	12 (15)	1.000	0.92 (0.05–16.4)
Never	1 (2.5)	1 (1.25)		Ref
**Raw vegetables**				
At least weekly	18 (45)	62 (77.5)	0.297	0.47 (0.14–1.60)
Less than weekly	17 (42.5)	10 (69.6)	0.185	0.37 (0.09–1.44)
Never	5 (12.5)	10 (11.4)		Ref
**Raw milk products**	(31)	(69)		
At least weekly	1 (3.2)	19 (27.6)	0.002	0.07 (0.01–0.56)
Less than weekly	4 (12.9)	15 (21.7)	0.109	0.36 (0.11–1.21)
Never	26 (83.9)	35 (50.7)		Ref
**Raw minced pork meat**	(37)	(78)		
At least weekly	2 (5.4)	2 (2.6)	0.598	2.07 (0.28–15.4)
Less than weekly	6 (16.2)	16 (20.5)	0.799	0.78 (0.28–2.19)
Never	29 (78.4)	60 (76.9)		Ref
**Cooked minced pork meat**	(37)	(79)		
At least weekly	5 (13.5)	5 (6.3)	0.205	3.75 (0.71–19.7)
Less than weekly	28 (75.7)	59 (75.6)	0.417	1.78 (0.54–5.86)
Never	4 (10.8)	15 (19.2)		Ref
**Pork steak**	(39)	(79)		
At least weekly	12 (30.8)	8 (10.1)	0.011	6.38 (1.56–26.1)
Less than weekly	23 (59.0)	54 (68.4)	0.415	0.55 (0.17–1.82)
Never	4 (10.2)	17 (20.5)		Ref
**Raw minced beef meat**	(38)	(77)		
At least weekly	1 (2.6)	2 (2.6)	1.000	1.00 (0.09–11.45)
Less than weekly	5 (13.2)	11 (14.3)	1.000	0.91 (0.29–2.84)
Never	32 (84.2)	64 (83.1)		Ref
**Beef steak undercooked**	(36)	(77)		
At least weekly	1 (2.8)	3 (3.9)	1.000	0.73 (0.07–7.36)
Less than weekly	9 (25)	17 (22.1)	0.812	1.16 (0.64–2.95)
Never	26 (72.2)	57 (74.0)		Ref
**Wild boar meat**	(37)	(58)		
Yes	2 (5.4)	7 (12.1)		
No	35 (94.6)	51 (87.9)	0.279	0.42 (0.08–2.12)
**Meat processing at home**	(38)	(70)		
At least weekly	12 (31.6)	3 (4.3)	0.027	5.43 (1.29–22.9)
Less than weekly	12 (31.6)	48 (68.6)	0.030	0.34 (0.13–0.87)
Never	14 (36.8)	19 (27.1)		Ref
**Touching raw meat with bare hands**	(22)	(64)		
Yes	18 (81.8)	37 (57.8)	0.043	3.28 (1.00–10.81)
No	4 (18.2)	27 (42.2)		Ref
**Salami, raw sausage**	(38)	(78)		
At least weekly	18 (47.4)	24 (30.8)	0.039	3.45 (1.10–10.83)
Less than weekly	15 (39.5)	31 (39.7)	0.189	2.23 (0.71–7.01)
Never	5 (13.1)	23 (29.5)		Ref
**Spreadable sausage (Teeschinken)**	(35)	(78)		
At least weekly	14 (40.0)	11 (14.1)	<0.001	7.16 (2.41–21.3)
Less than weekly	13 (37.1)	22 (28.2)		3.32 (1.20–9.20)
Never	8 (22.9)	45 (57.7)	0.023	Ref
**Raw ham**	(38)	(78)		
At least weekly	12 (31.6)	22 (28.2)	0.190	2.18 (0.77–6.21)
Less than weekly	18 (47.4)	24 (30.8)	0.034	3.0 (1.12–8.05)
Never	8 (21.1)	32 (41.0)		Ref
**Cured pork meat (Pökel)**	(35)	(77)		
At least weekly	5 (14.3)	14 (18.2)	0.015	7.14 (1.52–33.5)
Less than weekly	16 (45.7)		<0.001	4.90 (1.95–12.3)
Never	14 (40.0)	60 (77.9)		Ref
**Boiled sausage (Brühwurst)**	(36)	(77)		
At least weekly	12 (33.3)	14 (18.2)	0.002	5.88 (1.95–17.8)
Less than weekly	17 (47.2)	15 (19.5)	<0.001	7.77 (2.71–22.3)
Never	7 (19.4)	48 (62.3)		Ref
**Jellied sausage (Sülzwurst)**	(34)	(77)		
At least weekly	0 (0)	5 (6.5)	0.318	0.71 (0.62–0.81)
Less than weekly	9 (26.5)	10 (13.0)	0.173	2.23 (0.81–6.15)
Never	25 (73.5)	62 (80.5)		Ref
**Blood sausage (Blutwurst)**	(35)	(72)		
At least weekly	0 (0)	2 (2.8)	1.000	0.72 (0.63–0.82)
Less than weekly	10 (28.6)	6 (8.3)	0.010	4.27 (1.40–12.98)
Never	25 (71.4)	64 (88.9)		Ref
**Liver sausage (Leberwurst)**	(36)	(78)		
At least weekly	7 (19.5)	17 (21.8)	1.000	0.99 (0.33–2.94)
Less than weekly	16 (44.4)	29 (37.2)	0.654	1.36 (0.56–3.30)
Never	13 (36.1)	32 (41.0)		Ref
**Wiener sausage**	(36)	(77)		
At least weekly	8 (22.2)	22 (28.6)	0.440	0.64 (0.22–1.85)
Less than weekly	15 (41.7)	32 (41.6)	0.815	1.21 (0.48–3.01)
Never	13 (36.1)	23 (29.8)		Ref

Data are expressed as mean (±standard deviation) unless stated otherwise. Numbers in () depict the rate of answers given, omitting the answer “unknown” or missing values. ORs with multiple values were calculated compared with the answer “never” as the reference (Ref) value.

## Data Availability

The data presented in this study are available on request from the corresponding author. The data are not publicly available.

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
