# Peer review of "Risk Factors for Hepatitis E Virus Infection and Eating Habits in Kidney Transplant Recipients"

_pathogens, 2023, doi:10.3390/pathogens12060850_

Round 1

Reviewer 1 Report

Wu and colleagues investigated risk factors for HEV and then chronic HEV in a monocentric, retrospective cohort of kidney transplant recipients.

The study is overall well written but I have several concerns that need to be addressed before publication

1) Method:

_the authors aimed to investigate risk factor for HEV, including dietary. They included 59 patients with de novo post transplant HEV

However only 40 patients were included in the dietary survey. This is unclear why the survey wasn't performed for other patients.

_To investigate drug and demographics (i.e other than food habits) risk factors the authors used a cohort of kidney transplant patients with liver disturbance, but HEV RNA negative. I think this group is well adapted to compare with the HEV RNA+ group. However, concerning dietary habitus it seems that the comparative group was derived from another group, not really described in the manuscript. This is hard to follow for the reader.

Results:

_ in the manuscript the authors rotate risk factor for HEV and risk factor for chronic HEV. This need to be improved

_ the authors did not mentionned the real dose of ribavirine that they used to treat patients. This is a major limitation to draw conclusions

_ All tables need editing

_ The discussion need to be restructured:

section treatment of rejection, plasmapheresis : out of the scope, speculative and not data-driven. Specific discussion regarding potential useful actions to improve the high level of post transplant acute and chronic HEV should be detailled in a more precise fashion 

Author Response

We are truly thankful for the reviewer´s positive and helpful comments.

Here is our point-by-point response.

1) Method:

_the authors aimed to investigate risk factor for HEV, including dietary. They included 59 patients with de novo post transplant HEV

However only 40 patients were included in the dietary survey. This is unclear why the survey wasn't performed for other patients.

_To investigate drug and demographics (i.e other than food habits) risk factors the authors used a cohort of kidney transplant patients with liver disturbance, but HEV RNA negative. I think this group is well adapted to compare with the HEV RNA+ group. However, concerning dietary habitus it seems that the comparative group was derived from another group, not really described in the manuscript. This is hard to follow for the reader.

Response: We apologize that we missed this information within the manuscript. We analyzed all available dietary surveys of patients with post-transplant HEV. However, not every patient answered to the questions and/or not every treating physician asked about the patient´s eating habits during outpatient visits. In addition, regarding the retrospective nature of the study, patients lost to follow up many years after diagnosis of HEV could not take part in the survey. At the end, we had a number of n=40.

Regarding the control group, we realized a much lower rate of eating habit evaluations by the treating physician and/or less responders to the survey. Since all KTR were treated at our center, we completed dietary surveys during follow up visits in KTR without evidence of HEV infection. We now added this missing information within the method and result section: 

“Dietary surveys were available from 40 out of 59 patients with HEV infection. To compare eating habits with a control group of KTR, the same questionnaire was performed by 80 control KTR from our center (to reach a 1:2 ratio) with no evidence for HEV infection during the same observation period. Most patients were not part of the original control group since, due to the retrospective nature of the analysis, not enough dietary information was available for KTR patients without evidence of HEV infection.”

_ in the manuscript the authors rotate risk factor for HEV and risk factor for chronic HEV. This need to be improved

Response: We improved our wording and uniformly changed the term into risk factor for HEV.

_ the authors did not mentionned the real dose of ribavirine that they used to treat patients. This is a major limitation to draw conclusions

Response: Thank you for your helpful comment. We looked into the data and created a new Table.

We analyzed the following aspects: what was the median ribavirin start dose between KTR with treatment success and treatment failure? Was the starting dose below the recommended GFR-adapted dose? How often was the dose increased or reduced during treatment? We also compared rates of EPO treatment before and after start of RBV.

We observed the following results:

Mean RBV doses in both groups did not differ, but we questioned, if the start dose was inadequately low or if RBV-induced side effects, especially anemia, lead to interruption of RBV treatment, dose reduction and, as a consequence higher rates of viral relapse. Detailed characteristics are depicted in the new Table 5. We observed a lower median RBV dose at initiation of treatment in patients with RBV treatment failure versus patients with successful viral clearance (300 versus 400 mg). Regarding the high rate of relevant anemia in both cohorts (78.9 and 58.6%), rates of EPO treatment and start of EPO treatment during RBV treatment (73.6 and 55.1%) were not significantly higher in patients with RBV treatment failure, but there was a significant delay of start with EPO substitution during treatment in patients with treatment failure (after 67.6 ± 49.5 versus 16.0 ± 15.2 days), possibly relevant for higher rates of anemia and interruptions of RBV treatment and/or dose reductions.

We added these findings within the result and discussion section.

_ All tables need editing

Response: we apologize. We realized that especially row 1 within all tables changed after the submission process. We re-edited all tables to improve reading, but we guess, that final editing will be done by the journal.

_ The discussion need to be restructured:

section treatment of rejection, plasmapheresis : out of the scope, speculative and not data-driven. Specific discussion regarding potential useful actions to improve the high level of post transplant acute and chronic HEV should be detailled in a more precise fashion 

Response: We agree and removed the section discussing treatment or rejection. Moreover we added potential useful actions to prevent post-transplant HEV infection or treat KTR with HEV infection more efficiently. Changes within the discussion are highlighted in red.

Reviewer 2 Report

      I.          The study is well-designed, maximum clinical parameters are included concerning KTR and food associated with enzyme and protein metabolism concerning HEV and RBV, with interesting parameters significantly associated with each other. The work is highly appreciated. With some more effort and statistical analysis some other significant markers can be extracted from the data, hence recommendations are as below, please.

    II.          The author has mentioned this in Table 1. Baseline characteristics of KTR with elevated liver enzymes by status for HEV infection. Triple IS (please write detail which three IS) and Dual IS (please write detail which two IS).

  III.          How the various clinical parameters performed in patients who were treated with cyclosporine A (52 vs 222 / p-valve 0.088). It is recommended to present the data. Keep the cyclosporine as a constant variable and against other clinical variables.

  IV.          Rise of proteinuria at liver enzyme elevation, 15/50 (30.0), 31/215 (14.4), 0.009, at this place, ALT and AST p-value is recommended concerning the increase of proteinuria, which enzyme is highly more associated with the increase of proteinuria?

    V.          A table is recommended, of Baseline characteristics of KTR with elevated creatinine by status for HEV infection, for detail see the p-values (0.046, 0.014, 0.008, and 0.086) of eGFR during HEV infection stable. deterioration (≥5 mL/min), Baseline proteinuria (mg/g creatinine), Peak proteinuria during HEV infection (mg/g creatinine), and Rise of proteinuria during HEV viremia (≥300 mg/g creatinine)

  VI.          Please share the possible explanation of Table 4. Characteristics of RBV treated KTR by treatment outcome, Duration of RBV treatment (d) refer 471 /118; <0.001, as why the patient fails to respond to RBV treatment? (Age, BMI, or?)

VII.          A table is recommended: Predictors of KTR, which are highly associated (like, creatine, IS, BMI, RBV, AST, ALT, IgG/ IgM )  multivariable analysis.

Please also consult the reference, 

Sakulsaengprapha V, Wasuwanich P, Thawillarp S, Ingviya T, Phimphilai P, Sue PK, Jackson AM, Kraus ES, Teshale EH, Kamili S, Karnsakul W. Risk factors associated with Hepatitis E virus infection in kidney transplant recipients in a single tertiary Center in the United States. Transpl Immunol. 2023 Jun;78:101809. doi: 10.1016/j.trim.2023.101809. Epub 2023 Mar 1. PMID: 36863665.

ORCID numbers are missing? 

Author Response

I.  The study is well-designed, maximum clinical parameters are included concerning KTR and food associated with enzyme and protein metabolism concerning HEV and RBV, with interesting parameters significantly associated with each other. The work is highly appreciated. With some more effort and statistical analysis some other significant markers can be extracted from the data, hence recommendations are as below, please.

Response: We are truly thankful for the reviewer´s positive and helpful comments.

Here is our point-by-point response.

II. The author has mentioned this in Table 1. Baseline characteristics of KTR with elevated liver enzymes by status for HEV infection. Triple IS (please write detail which three IS) and Dual IS (please write detail which two IS).

Response: We added more information regarding triple and dual IS within the text.

“The frequency of triple und dual maintenance immunosuppression was similar in both groups. Triple immunosuppression mainly consisted of a combination of CNI, MPA and low dose corticosteroid (36 out of 45 in KTR with HEV infection and 160 out of 176 in the CG). Dual maintenance immunosuppression mainly consisted of a combination of CNI and MPA, which was more often than combining CNI and corticosteroid or combinations with mTOR inhibitor or belatacept.”

  III.          How the various clinical parameters performed in patients who were treated with cyclosporine A (52 vs 222 / p-valve 0.088). It is recommended to present the data. Keep the cyclosporine as a constant variable and against other clinical variables.

Response: Regarding the significant differences in use of cyclosporine in Table 1 (3 vs 52 /p-value 0.04), we performed another multivariable analysis to adjust for confounding. In a stepwise approach we analyzed cyclosporine, tacrolimus, gender, use of rituximab, use of high dose steroids and thymoglobulin. We realized that use of cyclosporine instead of tacrolimus correlated significantly with lower risk of HEV infection (0.24 (0.07-0.80)). We believe that intensity of immunosuppression matters and that tacrolimus instead of cyclosporine might predispose to more viral infections, including HEV infection. We therefore changed Table 2 to add more results of the multivariable analysis. To address the reviewer´s question to characterize patients with CyA intake, we added a new supplementary Table 1 with clinical parameters for KTR with CyA intake versus no CyA intake (and tacrolimus intake instead in most of the cases).

From our clinical perspective, cyclosporine as the first available CNI inhibitor was used for maintenance therapy in earlier times and was kept in case of stable allograft function. Second, switch from tacrolimus to CyA was used as an alternative option in case of tacrolimus-induced side effects or recurrent episodes of infection. In contrast, KTR with allograft rejection episodes were mainly treated with tacrolimus due to stronger immunosuppressive effects. Our new analysis showed that use of CyA was not associated with gender, age, co-morbidities or type of transplant.  We observed a significant longer time since transplantation in KTR with CyA intake. Furthermore, CyA was used less often in combination of immunosuppressive triple therapy regimens. We now added the new Table as a supplementary Table 1 to the manuscript, described the results within the result and discussion section.

IV. Rise of proteinuria at liver enzyme elevation, 15/50 (30.0), 31/215 (14.4), 0.009, at this place, ALT and AST p-value is recommended concerning the increase of proteinuria, which enzyme is highly more associated with the increase of proteinuria?

Response: We tested rise of ALT and AST with rise of proteinuria using Chi square analysis, but did not observe a correlation between rise of liver enzymes and rise of proteinuria, thus kidney damage is likely due to other reasons than enzyme elevation by itself. We realized that the term “Rise of proteinuria at liver enzyme elevation” in Table 1 is somehow misleading. We analyzed rise of proteinuria during the time period of HEV infection or, in the control group, during the same observation period. Thus, we changed the term now into “Rise of proteinuria during observation period” to address both cohorts, patients with HEV infection and the control group.

V.  A table is recommended, of Baseline characteristics of KTR with elevated creatinine by status for HEV infection, for detail see the p-values (0.046, 0.014, 0.008, and 0.086) of eGFR during HEV infection stable. deterioration (≥5 mL/min), Baseline proteinuria (mg/g creatinine), Peak proteinuria during HEV infection (mg/g creatinine), and Rise of proteinuria during HEV viremia (≥300 mg/g creatinine)

Response: The reviewer asked, if KTR with impaired baseline kidney function were the main reason for deterioration of GFR and rise of proteinuria during our observation period. We therefore divided KTR by status for impaired allograft function (defined as creatinine >1.4 mg/dL, to keep in mind the fact of a single kidney transplant) with detailed information regarding baseline creatinine, eGFR and proteinuria. We analyzed the outcome parameters peak proteinuria, rise of proteinuria and stable versus deterioration of eGFR during HEV infection. Result are shown in the supplementary Table 2.

While baseline proteinuria did not differ between both groups, we observed a higher peak proteinuria during HEV infection in the KTR group with impaired allograft function (p=0.042). In contrast, the percentage of KTR with rise of proteinuria (>300 mg/g Crea) during HEV infection was not higher in KTR with elevated creatinine (27.2% versus 23.1%). Furthermore, numbers of stable eGFR were higher in KTR with creatinine ≤1.4 mg/dl (50% versus 39.4%), but without significance (p=0.441). Thus, we conclude, that increases of proteinuria and deterioration of kidney function during HEV infection do not solely relate to chronic damages of the kidney transplant, but were potentially HEV-associated.

We now added this supplementary Table 2 to the manuscript and describe these details within the result and discussion section.

VI. Please share the possible explanation of Table 4. Characteristics of RBV treated KTR by treatment outcome, Duration of RBV treatment (d) refer 471 /118; <0.001, as why the patient fails to respond to RBV treatment? (Age, BMI, or?)

VII.          A table is recommended: Predictors of KTR, which are highly associated (like, creatine, IS, BMI, RBV, AST, ALT, IgG/ IgM )  multivariable analysis.

Response to point VI and VII: RBV treatment failure implied prolonged treatment duration to aim to reach viral clearance. Patients with older age (>60y) and lower BMI failed to reach viral clearance as shown by multivariable analysis in Table 5. We also tested all parameters with a p-value <0.200, but did not find significant predictors. If not significant, parameters were not shown.

We considered the possibility of a lower ribavirin start dose in patients with RBV treatment failure. A potentially insufficient RBV dose at start of treatment might lead to relapses as a consequence. Reasons for treatment failure and less rates of SVR might be due to relevant RBV side effects and anemia. Therefore, we analyzed the following aspects: what was the median Ribavirin start dose? Was the starting dose below the recommended GFR-adapted dose? How often was the dose increased or reduced during treatment? We also compared rates of EPO treatment before and after start of RBV.

We observed the following results: the median RBV dose at initiation of treatment was lower in patients with RBV treatment failure versus patients with successful clearance (300 versus 400 mg) and RBV dose was increased during treatment after a mean time 204 +/- 194 days explaining allover similar doses in both groups. Regarding the high rate of relevant anemia in both cohorts (78.9 and 58.6%), EPO treatment and start of EPO treatment during RBV treatment (73.6 and 55.1%) was not significantly higher in patients with RBV treatment failure, but there was a significant delay in patients with treatment failure (after 67.6 ± 49.5 versus 16.0 ± 15.2 days), which might be relevant for higher rates of anemia and interruptions of RBV treatment and/or dose reductions (see the new Table 5).

Finally, we discussed the higher frequency of RBV mutations in patients with treatment failure.

Please also consult the reference, 

Sakulsaengprapha V, Wasuwanich P, Thawillarp S, Ingviya T, Phimphilai P, Sue PK, Jackson AM, Kraus ES, Teshale EH, Kamili S, Karnsakul W. Risk factors associated with Hepatitis E virus infection in kidney transplant recipients in a single tertiary Center in the United States. Transpl Immunol. 2023 Jun;78:101809. doi: 10.1016/j.trim.2023.101809. Epub 2023 Mar 1. PMID: 36863665.

Response: We added the reference within the discussion section.

Round 2

Reviewer 1 Report

The authors have accurately responded to all of my concerns
Information presented is clear and meaningful for clinicians

No additional comments

Reviewer 2 Report

Comments reply is appreciated. 

Please, have look on Author's names spell, affiliation, and contribution.